# Characterisation of the Viral Community Associated with the Alfalfa Weevil (*Hypera postica*) and Its Host Plant, Alfalfa (*Medicago sativa*)

**DOI:** 10.3390/v13050791

**Published:** 2021-04-28

**Authors:** Sarah François, Aymeric Antoine-Lorquin, Maximilien Kulikowski, Marie Frayssinet, Denis Filloux, Emmanuel Fernandez, Philippe Roumagnac, Rémy Froissart, Mylène Ogliastro

**Affiliations:** 1Peter Medawar Building for Pathogen Research, Department of Zoology, University of Oxford, South Park Road, Oxford OX1 3SY, UK; 2DGIMI Diversity, Genomes and Microorganisms–Insects Interactions, University of Montpellier, INRAE, 34095 Montpellier, France; aymeric.antoine-lorquin@inrae.fr (A.A.-L.); kulikowski.max@live.fr (M.K.); marie.frayssinet@inrae.fr (M.F.); 3CIRAD, UMR PHIM, 34090 Montpellier, France; Denis.Filloux@Cirad.Fr (D.F.); Emmanuel.fernandez@cirad.com (E.F.); philippe.roumagnac@cirad.fr (P.R.); 4PHIM Plant Health Institute, University of Montpellier, CIRAD, INRAE, Institut Agro, IRD, 34090 Montpellier, France; 5MIVEGEC Infectious and Vector Diseases: Ecology, Genetics, Evolution and Control, University of Montpellier, CNRS, IRD, 34394 Montpellier, France; remy.froissart@montp.cnrs.fr

**Keywords:** insect pest, agroecosystem, viral metagenomics, virus diversity, biocontrol

## Abstract

Advances in viral metagenomics have paved the way of virus discovery by making the exploration of viruses in any ecosystem possible. Applied to agroecosystems, such an approach opens new possibilities to explore how viruses circulate between insects and plants, which may help to optimise their management. It could also lead to identifying novel entomopathogenic viral resources potentially suitable for biocontrol strategies. We sampled the larvae of a natural population of alfalfa weevils (*Hypera postica*), a major herbivorous pest feeding on legumes, and its host plant alfalfa (*Medicago sativa*). Insect and plant samples were collected from a crop field and an adjacent meadow. We characterised the diversity and abundance of viruses associated with weevils and alfalfa, and described nine putative new virus species, including four associated with alfalfa and five with weevils. In addition, we found that trophic accumulation may result in a higher diversity of plant viruses in phytophagous pests compared to host plants.

## 1. Introduction

Our knowledge of viruses infecting insects is largely skewed towards a small number of species of economical or health concerns in humans and their domesticated plants and animals [1]. Yet, insects belong to the most widespread and diversified group of animals on Earth, and despite their relatively small size, they represent a significant part of the terrestrial biomass [2]. In this context, and assuming that virus diversity correlates with the diversity of their hosts, insects could represent an extraordinary reservoir and a potential “nursery” of virus diversity [3].

Viral metagenomics has revolutionised virus discovery from samples of any nature and at all scales, i.e., from the scale of an individual to populations living in the same environment [4,5,6,7,8]. Nowadays, a viral metagenomic approach combined with high-throughput sequencing and dedicated bioinformatic tools allows the exploration of a large fraction of the diversity of viruses with no a priori knowledge. This approach also allowed the “pathogen-centered” approach relying on virus amplification that had long prevailed for virus discovery to be overcome [9]. Viral metagenomics can be now applied to the surveillance of agro-ecosystems (agrosystems), i.e., anthropized ecosystems dedicated to food production, and can help to understand how viruses, including pathogenic ones, circulate and impact their functioning [10].

Insects are integral components of agrosystems, where they fulfil crucial functions, if only to cite pollination. However, the abundance of food and favourable trophic conditions can make some herbivorous or sap-feeding species switch to real “pests of concern”. Management plans to reduce insect pests’ pressure on food production have long relied on chemical insecticides, which are easy to use and efficient, making made them success [11]. As a counterpart, chemicals have rapidly induced resistances in pest populations, and their lack of specificity has triggered overall stresses on ecosystems stability and resilience capacity including detrimental effects on human health and on biodiversity [12]. All these reasons make the reduction in their use a great challenge for future decades and push policy makers to expand and diversify alternative methods to control pests.

Biocontrol refers to the use of natural enemies to control pests, among which microbial pesticides and viruses have proven efficient and successful, although their use remains limited to a few cropping systems [13]. Entomopathogenic viruses may represent a largely unexplored biocontrol resource [13], so exploring agrosystems using metagenomics may help to understand the diversity and structure of virus communities associated with insect pests. Such exploration may also help to find local, adapted viral solutions and design adapted management strategies for the sustainable control of insect pests.

The weevil *Hypera postica* (Coleoptera, Curculionidae) is a major widespread pest that can reach high densities on leguminous plants. This pest particularly affects alfalfa (*Medicago sativa* L.) yield production, resulting in significant economic losses in Europe and the USA [14]. Damages to alfalfa plants are first caused by larvae, although adults feeding on alfalfa buds significantly affect the plant regrowth, which worsens the reduction in dry matter production [15]. Chemical management remains the main method to control *H. postica*, although few biocontrol trials with parasitic wasps (*Bathyplectes* species), generalist predators (coccinalids) and fungal pathogens have been reported [16,17]. However, because our understanding of viruses infecting curculionids is scarce, only one entomopathogenic virus is used to control these beetles to our knowledge: an iridovirus (Chilo iridescent virus) tested to control the cotton weevil (*Anthonomus grandis*) [18].

The objective of this work was to explore the diversity of viruses associated with the alfalfa weevil using a viral metagenomic approach. We characterized for the first time the diversity of viruses that can be found in this pest, including four new putative virus species infecting alfalfa and five new virus species infecting weevils.

## 2. Materials and Methods

### 2.1. Sampling Alfalfa and Alfalfa Weevils

Larvae of alfalfa weevils (*H. postica*) and alfalfa leaves (*M. sativa*) species were collected on the 1st April 2015 in Domaine de Restinclières, Prades-le-Lez, France (43.705043 °N, 3.863410 °W). The sampling took place in an experimental plot including an alfalfa field that did not receive any insecticide and an adjacent meadow. Insects were collected with nets on a surface of 100 m^2^/sample in both crop and meadow. Alfalfa leaves were randomly collected from plants located in the crop field and in the neighbouring meadow. One leaf was sampled per individual. Samples were maintained and transported on ice to the lab and stored at −80 °C until processing. Appendix A summarizes all the samples and their associated metadata.

### 2.2. Insect Identification

Alfalfa weevils were identified morphologically and molecularly by PCR barcoding using the COI gene primers Uni-MinibarF1: 5′-TCC ACT AAT CAC AAR GAT ATT GGT AC-3′ and Uni-MinibarR1: 5′- GAA AAT CAT AAT GAA GGC ATG AGC-3′ [19]. Insect DNA was extracted from 10 grinded individuals using a Wizard DNA purification kit (Promega, Madison, WI, USA). PCR amplification was performed using a HotStarTaq Master Mix Kit (Qiagen, Hilden, Germany) according to the manufacturer’s protocol. The following cycling conditions were used: one cycle of 95 °C for 2 min, 5 cycles of 95 °C for 1 min, 46 °C for 1 min, 72 °C for 30 s and 35 cycles of 95 °C for 1 min, 53 °C for 1 min and 72 °C for 30 s. An additional final extension was then performed for 5 min at 72 °C. The yield of the PCR products was verified by the migration of 8 µl of PCR products to 2% agarose gel and visualised by staining with ethidium bromide. Amplicons were cleaned up using a Wizard SV Gel and PCR Clean-Up kit (Promega) according to the manufacturer′s protocol. Purified PCR products were sequenced by Sanger sequencing (Beckman Coulter Genomics, France). Chromatograms were checked for disparities between the forward and reverse reads, and consensus sequences for each amplicon, then taxonomic assignment was performed using BLASTn [20] against the BOLD database [21].

### 2.3. Sample Preparation and Sequencing

Viromes were obtained from 14 alfalfa weevil samples (each sample being a pool of individuals weighting from 250 to 1900 mg and containing between 50 and 500 larvae) and 2 alfalfa plant samples (each sample being a pool of leaves from 10 individuals) as described previously [22] (Appendix A). Briefly, approximately one gram of insect or plant material was processed using a virion-associated nucleic acid (VANA) based metagenomics approach to screen for the presence of DNA and RNA viruses. Insect samples were grounded and centrifuged to recover supernatants that were next filtered through a 0.45 µm filter and centrifuged at 140,000× *g* for 2.5 h to concentrate viral particles. The resulting pellets were resuspended, and nucleic acids not protected in virus-like particles (VLPs) were degraded by DNase and RNase incubation at 37 °C for 1.5 h. Total RNA and DNA were then extracted using a NucleoSpin kit (Macherey Nagel, Bethlehem, PA, USA). Reverse transcription was performed by SuperScript III reverse transcriptase (Invitrogen), cDNAs were purified by a QIAquick PCR Purification Kit (Qiagen, Hilden, Germany) and complementary strands synthesised by Klenow DNA polymerase I. Double-stranded DNA was amplified by random PCR amplification. Samples were barcoded during reverse transcription and PCR steps using homemade 26-nt Dodeca Linkers and PCR multiplex identifier primers. PCR products were purified using NucleoSpin gel and PCR clean-up (Macherey Nagel, Bethlehem, PA, USA). Three negative controls were added during (1) viral purification, (2) nucleic acid extraction, and (3) cDNA purification. Finally, libraries were prepared from purified amplicons and sequenced on an Illumina MiSeq V3 chemistry to generate 300 nt paired-end (PE) reads (Genewiz, South Plainfield, NJ, USA).

### 2.4. Viral Genome Reconstruction

Illumina adaptors were removed, and reads were filtered for quality (q30 quality and read length >45 nt) using cutadapt 1.18 [23]. Cleaned reads were assembled de novo into contigs using MEGAHIT 1.2.8 [24]. Taxonomic assignment was achieved on contigs of length >900 nt through searches against the NCBI gbvrl viral database using DIAMOND 0.9.22 with an e-value cutoff of <10^−5^ [25]. All contigs that matched virus sequences were selected and used as queries to perform reciprocal searches on NCBI non-redundant protein sequence database with an e-value cutoff of <10^−3^ in order to eliminate likely false positives. Viral contigs completion and coverage was assessed by iterative mapping using BOWTIE2 2.3.4.3 with the options end−to-end and very-sensitive [26]. Putative Open Reading Frames (ORFs) were identified using ORF finder (length cutoff >300 nt) on Geneious prime 2021.0.3 [27]. In all subsequent analyses, we focused only on full or near full coding sequences (>90% of CDS) based on the alignments of genomes with their closest relatives combined with ORF completeness, thus discarding contigs with partial CDS. The viral isolates that belonged to already described species were reconstructed as follows: after mapping against the closest reference isolate deposited in the NCBI nucleotide database, consensus sequences were generated using samtools 1.2 [28]. Mutations were called at depth ≥5 if they differed from the reference isolate; otherwise, sites were kept as those of the reference isolate.

### 2.5. Virus Discovery and Taxonomic Assignment

To determine if viral contigs belonged to new species, their nucleotidic sequences or their predicted protein sequences were aligned and compared with the 10 closest related viral genomes found by similarity searches performed above using MAFFT v7.450 with the G-INS-i algorithm [29,30] or MUSCLE 3.8.425 (16 iterations) [31] using default settings. Alignment results were visualised using SDT 1.2 [32], and genomes were classified as new virus strain versus new virus species according to the species demarcation thresholds recommended within the online reports of the International Committee on Taxonomy of Virus (ICTV) (https://talk.ictvonline.org/ accessed on 21 March 2021).

### 2.6. Phylogenetic Analyses

Phylogenetic trees were built using maximum likelihood methods for all reconstructed genomes corresponding to new species in order to place them within the currently known viral diversity and to infer their possible host range. Representative sets of replication and capsid proteins, or polyproteins, were extracted from the NCBI GenBank non-redundant (nr) database for each taxonomic group in which the genomes were classified (analysis performed on the 22 January 2021). Amino acid sequences were aligned using MUSCLE 3.8.425 (16 iterations) with default settings [31]. Sequences that were not reliably aligned due to high amino acid divergence were removed and the dataset subsequently realigned. Phylogenetic trees were constructed in RAxML 8.2.11 [33] using the LG + I + G protein evolution model. Tree branch support was estimated using 1000 bootstrapped replicates, except for the *Iflaviridae* and *Alphaflexiviridae* family trees for which 100 bootstrapped replicates were used. Trees were mid-point rooted and visualised with FigTree 1.4 (http://tree.bio.ed.ac.uk/software%20/figtree/ accessed on 21 March 2021).

### 2.7. Screening for Virus Presence in Samples Using PCR

The presence of the most abundant virus species found in weevils (Hypera postica associated iflavirus 1) was confirmed in *Hypera postica* samples using specific PCR and Sanger sequencing. Purified DNA was screened for the presence of this virus using a specific primer pair targeting a 1127 nt long region of the polyprotein gene (8007F: 5′-GCT GGC TTT TCA GAC GGC TCT A-3′, 9134R: 5′-TGG ATT ACC GCT AGG CAT CCC A-3′). A PCR mix was prepared using the HotStarTaq Master Mix kit (Qiagen, Hilden, Germany) according to the manufacturer′s protocol. Amplification conditions were as follows: 95 °C for 2 min, then 35 cycles of 95 °C for 2 min, 55 °C for 1 min, 72 °C for 2 min and a final extension step of 72 °C for 5 min. Amplicons were Sanger sequenced (Beckman Coulter Genomics, Marseille, France), and sequences were aligned with the reference genome sequence.

### 2.8. Statistical Analyses

Statistical analyses were performed using R and RStudio 1.2.5019 software [34,35]. Data from contingency tables were standardised to allow inter-sample comparisons: taxonomic binning artefacts and potential inter-sample contamination were restricted by (1) applying an abundance threshold >1/10,000 reads/taxon/sample, and (2) for taxa that contaminated negative controls, by removing them from sample datasets where their abundance was equal or inferior to their abundance in controls. Inter-virus taxa standardisation was performed by dividing, for each virus taxa, the number of virus reads by the length of viral contigs (kb). Viral diversity accumulation curves of *H. postica* viromes were made using the Vegan package [36]. Differences in viromes richness and composition of *H. postica* and *M. sativa* were visualised using barplots and a heatmap.

## 3. Results

### 3.1. Virome Composition and Virus Share between Alfalfa Weevils and Alfalfa Plants

The taxonomic assignment of contigs obtained from 16 samples (14 insect samples, each sample being a pool of 50 to 500 larvae; two alfalfa plant samples, each sample being a pool of leaves from 10 individuals—see Appendix A) to viral families was highly variable among samples, ranging between 10% and 75% (average 41%) of the total number of cleaned reads (Figure 1). However, the sequencing depth was not sufficient to recover the entire viral communities as indicated by the rarefaction curve of 3/16 samples that did not reach the asymptote (Appendix A). This feature might be due to the stringent abundance threshold we used to eliminate inter-sample contamination and artefactual taxonomic assignments (see materials and methods), which probably contributed to discarding rare viral taxa. In addition, our sampling effort is not sufficient to recover the entire diversity of viruses that were circulating in the targeted population of alfalfa weevils (Figure 2).

Overall, we found 23 virus families associated with the alfalfa agrosystem (Figure 3), including plant or fungus-associated viruses belonging to 11 virus families (*Alphaflexiviridae*, *Amalgaviridae*, *Bromoviridae*, *Caulimoviridae*, *Endornaviridae*, *Geminiviridae*, *Luteoviridae*, *Partitiviridae*, *Secoviridae*, *Tymoviridae* and *Solemoviridae*) and arthropod-infecting virus belonging to nine families (*Birnaviridae*, *Iflaviridae*, *Mesoniviridae*, *Parvoviridae*, *Permutotetraviridae*, *Phenuiviridae*, *Qinviridae*, *Reoviridae* and *Sinhaliviridae*). Sequences belonging to three families of bacteriophages (*Microviridae*, *Myoviridae* and *Siphoviridae*) were also detected but at low abundance **(**Appendix A). Remarkably, all viromes were dominated by RNA viruses, both in terms of family richness and abundance of reads, classified in 18 virus families corresponding to (+) ssRNA viruses that infect arthropods and plants.

Regarding weevil samples, viral sequences corresponding to families of arthropod-infecting viruses were the most abundant, representing on average 76% of the reads (the remaining 24% correspond to putative plant/fungus-infecting viruses) (Appendix A). More precisely, the weevil viromes were dominated (in read occurrence and abundance) by viruses belonging to the *Iflaviridae* family (Figure 3). They represent high read counts in all the samples. By contrast, viruses belonging to other families of arthropod-infecting viruses, i.e., *Parvoviridae* (subfamily *Densovirinae*), *Permutotetraviridae*, *Sinhaliviridae*, *Phenuiviridae*, *Qinviridae*, *Reoviridae*, *Birnaviridae* and *Mesoniviridae*, were detected at low abundance and low frequency, in one or two samples only (Figure 3 and Figure 4). Plant/fungus-infecting viruses belonging to seven families were only detected in weevils (i.e., *Solemoviridae*, *Secoviridae*, *Luteoviridae*, *Geminiviridae*, *Endornaviridae*, *Caulimoviridae* and *Alphaflexiviridae*) (Figure 3, Appendix A**),** where they can represent high read counts, like members of the family *Partitiviridae* (>1000 reads), in five weevil samples (Figure 4 and Appendix A). On the other hand, plant/fungal viruses belonging to four families were found in alfalfa samples (*Amalgaviridae*, *Bromoviridae*, *Partitiviridae* and *Tymoviridae*); this result is likely to be due to the low number of collected plants (20 individuals) compared to collected arthropods. No arthropod-infecting virus sequences were found in the viromes of alfalfa plant samples. Interestingly, members of families *Partitiviridae* and *Alphaflexiviridae* were present in most, if not all, weevil samples, although at various abundances. Members of the family *Partitiviridae* were found to be also highly abundant in alfalfa samples, unlike *Alphaflexiviridae* family members that were not detected in alfalfa samples, which might be explained by their high and low prevalence in the agrosystem, respectively. Finally, viruses belonging to the family *Amalgaviridae* displayed higher read counts in the alfalfa collected from the meadow compared to the crop. The low abundance and presence of the members of this virus family in the weevils sampled in the crop further support that amalgaviruses displayed a low prevalence and/or cause covert infection in alfalfa (Figure 4).

### 3.2. Virus Discovery

We reconstructed 17 contigs from the 16 samples, ranging from 1712 to 12,255 nt and we obtained the full coding sequences for all but one, for which we recovered > 95% of its coding sequence (Appendix A). The depth of coverage across all contigs is high, having an average depth ranging from 56 to 28,625 reads, representing a total of 2433,637 mapped reads (Appendix A).

Eight of these contigs correspond to six virus species infecting alfalfa that have already been reported [37]. Their genomes share high levels of genome identity with their closest relatives (98.2% to 100% *n*t identity). These viruses likely represent new isolates belonging to six plant virus species (Appendix A), namely, *Alfalfa mosaic virus* (*Bromoviridae*, *Alfamovirus* genus), *Medicago sativa amalgavirus 1* (*Amalgaviridae*, *Amalgavirus*), *Medicago sativa alphapartitiviruses 1* and *2* (*Partitiviridae*, *Alphapartitivirus*), *Bean leafroll virus* (*Luteoviridae*, *Luteovirus*) and *Alfalfa virus F* (*Tymoviridae*, *Marafivirus*) [37].

By contrast, nine genomes displayed a low identity, at the amino acid level, with their closest relatives (from 36.6% to 57.2% aa identity in the most conserved proteins with their closest relatives; Appendix A). According to the ICTV species demarcation guideline, we propose to classify three of these genomes as novel virus species and tentatively named them Hypera postica-associated alphaflexivirus (abbreviated to HpaAV) (*Alphaflexiviridae*, *Platypuvirus*; species demarcation is <80% aa sequence identity in the capsid or polymerase proteins [38]) and Hypera postica-associated iflavirus 1 and 2 (HpaIV1 and HpaIV2) (*Iflaviridae*, *Iflavirus*; species demarcation is <90% aa sequence identity in the capsid proteins [39]) (Appendix A). HpaIV1 was highly abundant and present in all weevil samples. We designed specific primers, and we could confirm their presence in insect samples by PCR. Five virus sequences belong to virus families with no current species demarcation threshold (*Solemoviridae*, *Sinhaliviridae* and *Permutotetraviridae*) and could not be proposed for classification, although they likely represent new species given their low aa identity with their closest relatives (Appendix A). Finally, we found one contig displaying low sequence identity with RNA viruses that are not yet classified at the family level. We tentatively named this virus the Hypera postica-associated virus 1 (HpaV1) (Appendix A).

### 3.3. Virus Phylogenetic Analysis

We next built phylogenetic trees using maximum likelihood methods for these nine genomes in order to place them within currently known viral diversity and to explore their possible host range. First, we analysed HpaAV (*Alphaflexiviridae*, *Platypuvirus*) and three *Hypera postica*-associated sobemoviruses (*Solemoviridae*, *Sobemovirus*). The *Alphaflexiviridae* family phylogenetic tree indicates that HpaAV clustered with viruses isolated from plants (Figure 5), but there is no experimental proof that they are plant infecting. The host range of this virus family currently only includes plants and plant-infecting fungi [38]. The *Solemoviridae* family phylogenetic tree shows that the three *H. postica*-associated sobemoviruses cluster with groups of viruses isolated from thrips (Figure 6); these insects and beetles are known vectors of sobemoviruses [40]. These results suggest either that HpaAV- and *H. postica*-associated sobemoviruses could infect alfalfa and their presence in weevils is likely due to trophic contamination and/or that they might be transmitted by weevils. With the exception of the HpaAV that displayed high read counts in some weevil samples (e.g., Hp04, Hp10 and Hp11), these viruses displayed low abundance/absence in alfalfa, which could suggest that their infection level in the plant and/or their prevalence in the crop is probably low (Figure 3 and Figure 4).

We further analysed the phylogenetic position of the five novel genomes assigned to insect-infecting virus families. HpaIV1 and HpaIV2 are members of the family *Iflaviridae* (genus *Iflavirus*), which comprises viruses that only infect arthropods. Both iflaviruses clustered with viruses discovered in Coleoptera (*Aulacophora lewisii* and *Lampirys noctiluca*), their closest relative being Aulacophora lewisii iflavirus 1 that was isolated from a beetle species (Figure 7). Similarly, Hypera postica-associated permutotetravirus clustered in the family *Permutotetraviridae*, which is composed of two genera with two virus species infecting lepidopteras recognised to date by the ICTV (Figure 8). It is noteworthy that other currently unclassified permutotetraviruses have been isolated from flies (*Drosophila melanogaster*) and leafhoppers (*Scaphoideus titanus*), suggesting that the genetic diversity and the host range of this virus family are probably more diverse than currently recognised. Hypera postica-associated sinaivirus is placed at the base of the three species members of the *Sinaivirus* genus (*Sinhaliviridae*) currently reported to infect bees (family *Apidae*; Appendix A).

Finally, we found one taxon, HpaV1, clustering with unclassified (+) ssRNA viruses, whose closest relatives belong to the Picornavirales order. Like members in this virus order (e.g., families *Iflaviridae* and *Dicistroviridae*), HpaV1 encodes for a long single polyprotein (12,204 nt in length). The closest relatives of this virus are the Wuhan house centipede virus 3 (NC_033458) and the Diabrotica undecimpunctata virus 1 (MN646770), which have been isolated recently from two arthropod species, including the house centipede (*Scutigera coleoptrata*) and the spotted cucumber beetle (*Diabrotica undecimpunctata*). Since these viruses all infect arthropods, we speculate that HpaV1 likely infects *H. postica* (Appendix A).

## 4. Discussion

The aim of this study was to explore the diversity of viruses circulating between an insect pest and its host plant using viral metagenomics. We collected samples of the alfalfa weevil pest and its host plant, both from a crop and a nearby meadow. The viral sequences we found in weevil larvae represented arthropod viruses belonging to nine families and plant/fungal viruses belonging to 11 families, while only plant viruses belonging to four families were detected in alfalfa. Viral sequences were non-uniformly distributed across samples: some were highly abundant in most samples, while others displayed more viral richness but a low number of reads. Although the method we used supposedly enables a substantial viral enrichment in libraries compared to other viral metagenomic studies [41,42,43], several methodological caveats can be indicated. First, our sampling effort and the sequencing depth were not sufficient to recover a comprehensive representation of the diversity of viruses that circulates in the sampled population of alfalfa weevils. This assumption is supported by the virus family accumulation curve, expected to plateau when only rare virus families remain to be described. Here, the accumulation curve did not reach an asymptote, thus supporting that we did not capture the full virus diversity and that the sampling effort should be increased for this aim. However, we cannot exclude that the stringent abundance threshold we determined also eliminated rare virus families, also adding a bias in the analysis of viral diversity.

Second, it is noticeable that most of the virus sequences reported here correspond to small RNA viruses (16 out of 20), with viral particle ranging in size from 30 to 80 nm, except for the members of the *Alphaflexiviridae* and *Phenuiviridae* families that have larger viral particles. The majority of these virus families correspond to (+) ssRNA viruses. We found reads classified in only three families of DNA viruses, two belonging to virus families whose host range comprises plants (*Geminiviridae* and *Caulimoviridae*) and one to a family comprising arthropod-infecting viruses (*Parvoviridae*), although they represent low numbers of reads so we could not reconstruct any of their full coding sequences. We cannot exclude that the greater abundance of small viruses could be due to the sample preparation method, based on virus-like particle enrichment and involving filtration and ultracentrifugation steps that are known to bias virome composition toward small viruses [44,45]. Methodological biases must be kept in mind when analysing the richness of viral communities.

The results we obtained from this field screening allowed us to identify arthropod- and plant/fungus-infecting viruses that potentially correspond to nine novel virus species. Interestingly, arthropod viruses were more divergent from their closest relatives than the putative plant/fungus-infecting viruses that we found in weevils. Moreover, six plant-infecting virus species described in this paper belong to already known species. These findings illustrate the poor knowledge we have of the diversity of arthropod-infecting viruses including from “well known” insect pests that threaten major cropping systems [1,46]. The presence of arthropod-infecting viruses in weevil larvae does not provide evidence that these viruses are indeed infectious for weevils [47]. An argument against this would be the close relatedness of these viruses to other virus species isolated from beetles.

Two novel RNA virus species (i.e., HpaIV1 and HpaV1) were relatively abundant in most of the insect samples, suggestive of a covert or an overt infection. Interestingly, the alfalfa weevil larvae complete their whole development on the same plant. This biological feature would make virus horizontal transmission and spreading to the population poorly efficient, limited to the few individuals living on the same plant, which is an argument for persistent infections and virus vertical transmission. Indeed, iflaviruses can often result in covert, sublethal infections in their insect hosts [48], although some can be lethal [49]. Covert infections often rely on the vertical transmission of the pathogen, which makes its transmission depend on the reproduction of their host [50]. Many insect viruses have mixed modes of transmission, and switching from one mode to another may occur, depending on environmental stressors and host life strategies [51]. Integrating viruses with mixed transmission modes in pest management requires understanding the molecular mechanisms involved in the disease, including the impact of co-infection.

In addition to arthropod viruses, this dataset highlights that the weevil samples also contain a higher diversity, although a low abundance, of putative plant/fungal viruses compared to plant samples. The richness of plant/fungal viruses in weevils can be attributed to sampling bias, both in the number of samples from pooled individuals (14 weevils versus two plant samples) and the number of individuals per sample (5–10-fold more individuals in weevil samples than in plant samples). However, the higher richness of plant viruses in insects can also be explained by (1) trophic accumulation, as already reported in dragonflies [52], or by (2) their transmission using insect vectors, as already described in sobemoviruses [40]. If we assimilate the virome of one plant to one weevil (plausible considering the “sedentariness” of larvae), we can extrapolate that weevil samples (14 samples, each made of >50 individuals) correspond roughly to the screening of 700–1000 plant equivalents in the area. Thus, we can estimate that trophic accumulation could increase plant virus richness in weevils by a factor >5 compared to alfalfa sampling. Hence, sampling phytophagous insects, particularly vector species, can provide a better proxy of phytopathogenic virus richness in their host plant than sampling plants, and should be included in surveys of plant pathogenic viruses in crops [53,54,55].

Today, viral metagenomics and high-throughput sequencing allow us to fully integrate viruses into the dynamics of agrosystems. Such approach allows us to detect infections in insect communities that have largely been ignored to date, in particular, concerning insects of concern, such as pests and vectors. Major challenges remain ahead, both conceptual and methodological: first, in basic virology, to integrate co-infections and virus communities into the understanding of the functioning of agrosystems; second, in applied virology, to better integrate “beneficial” viruses into sustainable management programmes.

## Figures and Tables

**Figure 1 viruses-13-00791-f001:**
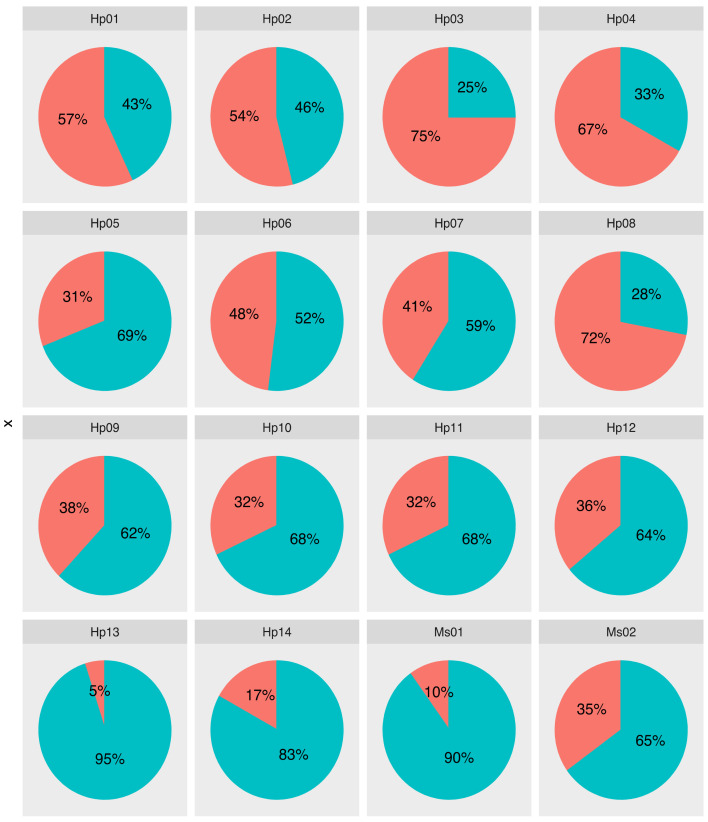
Proportion of reads classified (red) and unclassified (blue) at the family level. Hp: *Hypera postica* samples; Ms: *Medicago sativa* samples.

**Figure 2 viruses-13-00791-f002:**
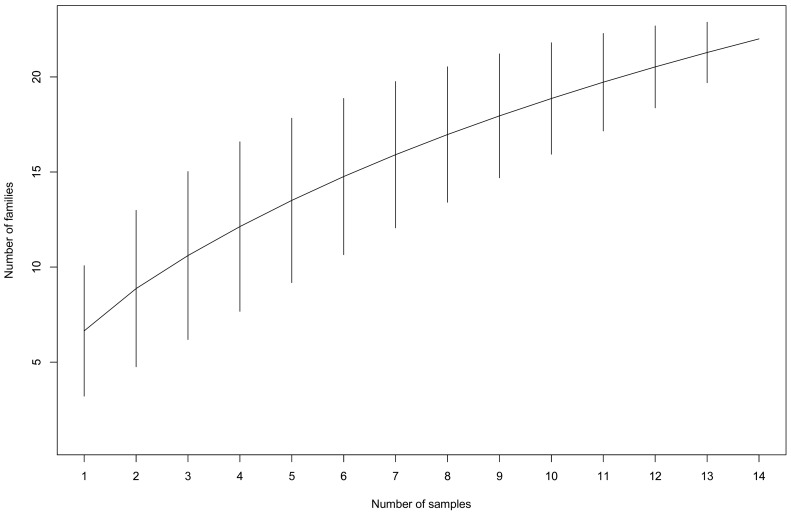
Accumulation curves of viral communities recovered from *Hypera postica* samples at the family level.

**Figure 3 viruses-13-00791-f003:**
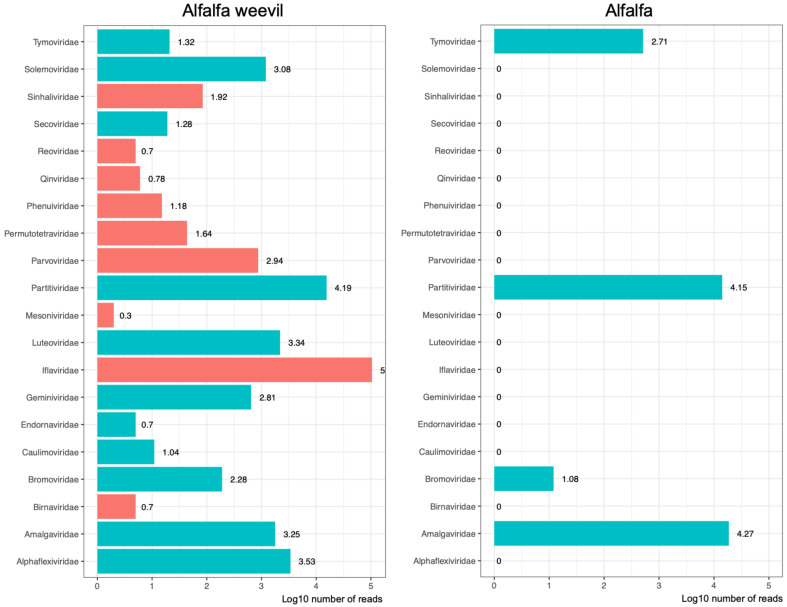
Abundance of arthropod-infecting viruses (red) and plant- or fungus-infecting viruses (blue) found in alfalfa weevils and in alfalfa. The horizontal axis represents the log10 number of reads attributed to each family.

**Figure 4 viruses-13-00791-f004:**
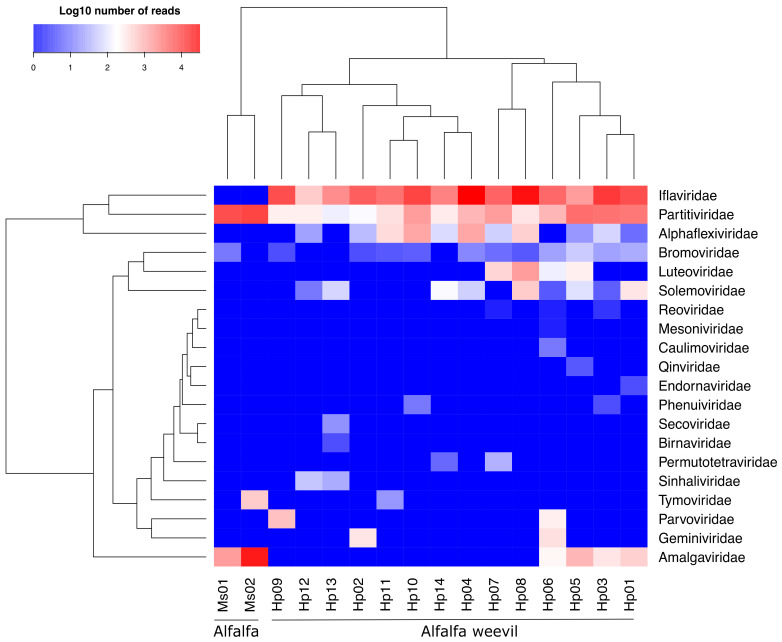
Heatmap representing the abundance of virus families in alfalfa (Ms) and alfalfa weevils (Hp). The colours represent the log10 number of reads attributed to each family.

**Figure 5 viruses-13-00791-f005:**
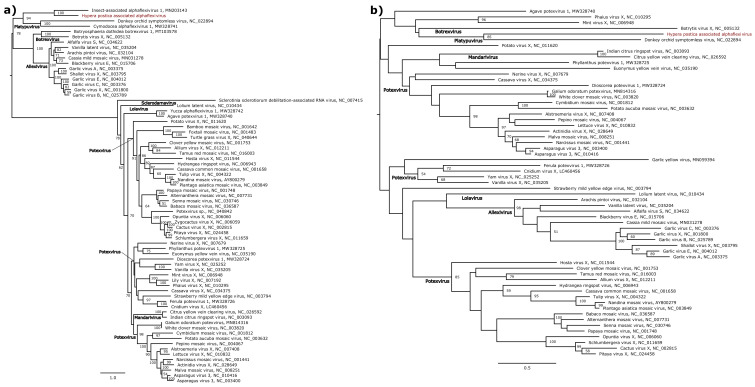
Maximum likelihood phylogenetic tree based on (**a**) the RNA-dependent RNA polymerase protein of 71 alphaflexiviruses, and (**b**) on the capsid protein of 61 alphaflexiviruses. The virus reported in this study is marked in red. The tree was mid-point rooted. Bootstrap values (100 replicates) superior to 50% are indicated at each node. Scale bar corresponds to amino acid substitutions per site.

**Figure 6 viruses-13-00791-f006:**
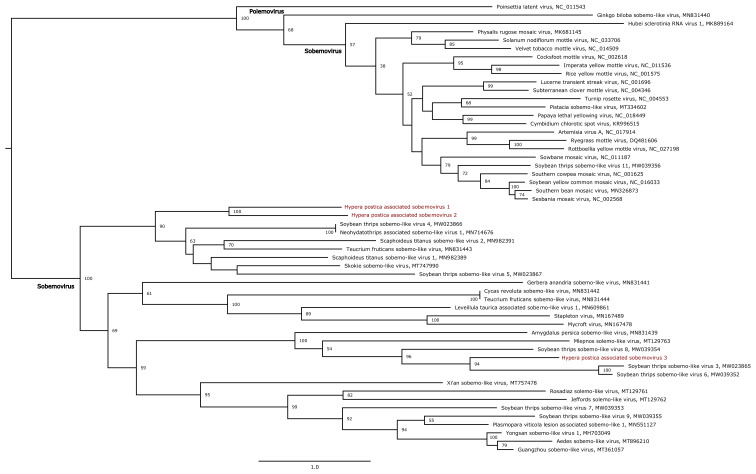
Maximum likelihood phylogenetic tree based on the RNA-dependent RNA polymerase protein of 54 sobemoviruses. Viruses reported in this study are marked in red. The tree was mid-point rooted. Bootstrap values (1000 replicates) superior to 50% are indicated at each node. Scale bar corresponds to amino acid substitutions per site.

**Figure 7 viruses-13-00791-f007:**
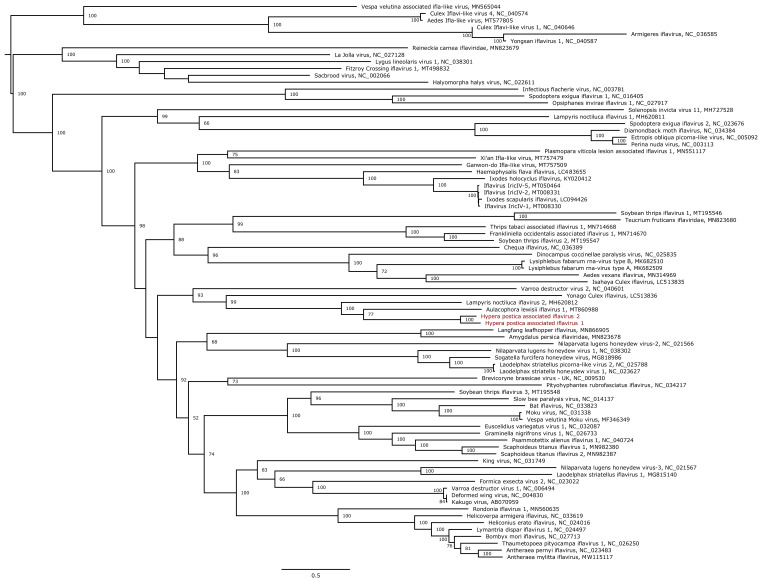
Maximum likelihood phylogenetic tree based on the polyprotein of 81 iflaviruses. Viruses reported in this study are marked in red. The tree was mid-point rooted. Bootstrap values (100 replicates) superior to 50% are indicated at each node. Scale bar corresponds to amino acid substitutions per site.

**Figure 8 viruses-13-00791-f008:**
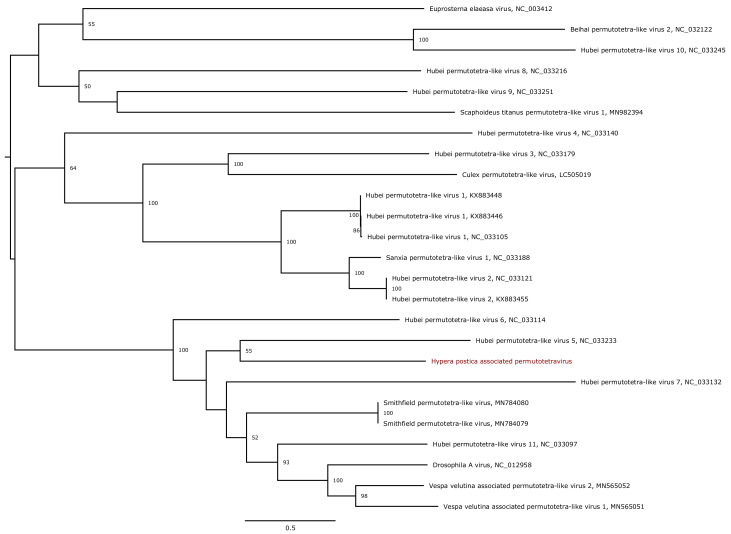
Maximum likelihood phylogenetic tree based on the RNA-dependent RNA polymerase protein of 25 permutotetraviruses. The virus reported in this study is marked in red. The tree was mid-point rooted. Bootstrap values (1000 replicates) superior to 50% are indicated at each node. Scale bar corresponds to amino acid substitutions per site.

## Data Availability

The reconstructed genomes were deposited in GenBank under the accession numbers MW676127 to MW676142. The metagenomic datasets used in this study were deposited on the SRA database with the accession numbers SRR13786451 to SRR13786466 under the BioProject PRJNA704818.

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
