# Peer review of "Characterisation of the Viral Community Associated with the Alfalfa Weevil (Hypera postica) and Its Host Plant, Alfalfa (Medicago sativa)"

_viruses, 2021, doi:10.3390/v13050791_

Round 1

Reviewer 1 Report

François et al. report in their manuscript a comparative metagenomic analysis of the viromes of the alfalfa weevil (Hypera postica) and its host (Medicago sativa), from samples collected in the same crop and surrounding area. As a result, the authors identified five novel putative viruses in the weevil and another four in alfalfa. This information is very valuable in order to identify possible sources for the “virocontrol” of the alfalfa weevil, a harmful species for which biological control is difficult.

Comments:

The authors mention the identification of a plausible members of DNA viruses plausibly belonging to families Geminiviridae, Caulimoviridae and Parvoviridae. This is a very interesting finding, but no more information is provided in the manuscript apart from that statement. Can the authors provide more details on this question?

I think that the Supplementary figure 2 should be moved to a figure in the main text, as it is clear indicative of the biological diversity of the viral taxonomic units in the samples and that show that the plateau has not been reached with 14 samples. As the authors explain, probably more unidentified viruses can be present in the samples not detected by the sequencing and bioinformatics pipeline or could happen that more samples need to be studied to be comprehensive enough of the virome of the alfalfa weevil in the alfalfa crop.

There are some misuses of terms throughout the text. For example, in L227 it reads: “only four plant/fungi infecting virus families were found in alfalfa samples”:  virus families do not infect plants or fungi, but the viruses themselves. Therefore, it would be more convenient to say that “… plant/fungal viruses belonging to the families ... were found in alfalfa samples”. Another example (L231): " The family Partitiviridae was found to be also highly abundant in alfalfa samples unlike Alphaflexiviridae”, but in fact the expression should be something like: "Members of the family Partitiviridae were found to be…”. Similar expressions are found throughout the text and need to be corrected.

Provide a figure for the RT-PCR detection of Hypera postica associated iflavirus 1 in the insect samples.

Provide captions and numbers for the Figures and Tables in the Supplementary material file, this would help the readability of the manuscript.

Propose provisional abbreviations for the all the new viruses identified: e.g. Hypera postica associated iflavirus 1 = HpaIV1, etc. and use along the text.

P5 L211: change “Noteworthy” to “Remarkably”

P6 L224: add “)” after “Alphaflexiviridae”

Reviewer 2 Report

The article by François et al. describes the diversity and abundance of viruses associated with the alfalfa weevil and its host plant, alfalfa. To achieve their goal, the authors attempted to explore what they call agrosystemic approach combined with metagenomics studies. The research is interesting and skillfully accomplished and the article is well-written.

 I think this report can be accepted for publication in Viruses, providing all of the following comments are thoroughly addressed and the article is revised to clarify and explain the deficiencies stated below:

  1. L22 and L75: Similarly to ecosystem, agrosystem represents an interacting community of living organisms and their physical environment. Since the research involved insects and plants only and no other environmental components were engaged, perhaps the all-embracing term “agrosystem” could be reconsidered and replaced with a more precise description. Especially bearing in mind a quite limited number of the samples.
  2. L86-87. Sampling: it appears that sample size that is, a number of replicates needed for a reliable observation, particularly in a metagenomic-based research (large scale study of environmental samples), is rather small.
  3. L87: ”and from ten individual plants growing in the neighboring meadow”. It is not clear if those plants collected in the meadow were also alfalfa plants.
  4. L109-110: “Viromes were obtained from 14 alfalfa weevil samples (each sample being a pool of individuals)”. Again, a sample size is quite limited to represent the entire “agrosystem”. Besides, how many individuals were pooled in one sample? At the end, the small sample size would translate into a reduced virus diversity.
  5. L110: “and 2 alfalfa plant samples”. Only two alfalfa samples? Or 20 leaves from 10 plants, as mentioned above on L86-87? Two samples are definitely not enough for any downstream conclusions. Please correct or clarify.
  6. L125-126: were the weevil samples and alfalfa samples loaded in separate lanes to ensure that there is no cross-contamination (despite of bar-coding)? Or were they multiplexed?
  7. L167-168: Please explain why only 100 bootstrapped replications were used for Iflaviridae and Alhpaflexiviridae. In this case, are confidence values on phylogenetic trees going to be robust enough?
  8. L194-195: “14 insect samples and 2 plant samples”. This affirms my worst suspicion that only two alfalfa samples were used in this study to estimate a virus diversity in the plant. I do not think this is acceptable. Please correct, clarify, or increase sample size.
  9. L198 and below: Perhaps I am missing it, but I do not see any figures numeration in the supplementary material to compare with legends and properly judge it.
  10. L201-201: “In addition, our sampling effort might not have been sufficient to recover the entire diversity of viruses that were circulating in the targeted agrosystem (supplementary figure 2).” Obviously, the authors themselves clearly understand pitfalls of this study.
  11. L227-228: likely because of only two alfalfa samples used in the study.
  12. L232-233: Does it mean that viruses of the Alphaflexiviridae family were still found in alfalfa? Why it is not mentioned above, on L227-228?
  13. L256-258, 267: according to the ICTV rules of orthography, only classified (accepted) species names can be italicized.
  14. L273-275: It is stated on L227-228 that “only four plant/fungi infecting virus families were found in alfalfa samples (Amalgaviridae, Bromoviridae, Partitiviridae and Tymoviridae).” It is therefore not clear why the authors suggest that hypera postica associated alphaflexivirus (Alphaflexiviridae) and three Hypera postica associated sobemoviruses (Solemoviridae)” are in fact infecting alfalfa and “their presence in weevils results from trophic accumulation”? The results should be uniformly clear: whether more than four families were found in alfalfa or these four viruses were only found in weevils. If the latter is true, the suggestion on “trophic contamination” would probably has to go.
  15. L278-279: Once again, if the viruses were not found in alfalfa, how could they contaminate and/or accumulate in weevils?
  16. Considering that only a handful of new genomes were analyzed, I am not entirely convinced that phylogenetic trees shown in Fig. 4-7 should include so many other known species to establish phylogenetic relationships.
  17. L341-343: “First, our sampling effort and the sequencing depth were probably not sufficient to recover a comprehensive representation of the diversity of viruses that circulates in this area.” Perhaps, the title, abstract and the body of the text should be slightly amended to accommodate this fact (insufficient number of samples) to a fuller degree by adding words/phrases such as ”limited, restricted, trial, or pilot” screening and somewhat downplaying expressions like “agrosystemic approach” and “fill the gap of knowledge on the diversity of viruses”, which are implying a large-scale testing.
  18. L364: If the authors mean four species that cluster with plant-infecting viruses (L273-275), was this experimentally proved that those are indeed plant-infecting? If not, perhaps they need to clarify that they refer to the “remaining 24% correspond to plant-infecting viruses”. Also, is there a possibility that some of those 24% of plant-infecting viruses found in weevils can be vectored by them?
  19. L364: on L205, the authors mention that they found “11 plant or fungus-associated viruses”
  20. L385-387 – the same (comment #18) is true for this sentence.
  21. L390-391: To my knowledge, none of these cited studies (51-53) specifically refers to “trophic accumulation”. Or uses this phenomena to explain virus diversity as suggested by the authors. This would have to be clarified.

Minor comments:

L40-41: “pathogen-centered”?

L51-51: “chemicals have rapidly selected resistances”. Perhaps the authors meant “induced”? This sentence could be improved or phrased differently.

L65: Medicago sativa L. ?

L71-72: “Regarding the current usage of entomopathogenic viruses, viruses…”. Perhaps, this sentence could be rephrased.

L78: alfalfa-infecting and weevils-infecting? Or: “four new virus species infecting alfalfa and five new viruses infecting weevils.”

Round 2

Reviewer 2 Report

In my view, the revised manuscript titled "Characterization of the viral community associated with the alfalfa weevil (Hypera postica) and its host plant, alfalfa (Medicago sativa)" by François et al., is acceptable for publication in Viruses as is, without further amendments.

Author Response

We thank again the reviewer for their insightful review.

We made one additional minor revision, as required by the editor:

On 2.1 you stated « Alfalfa leaves were randomly collected from ten
individual plants”, but later on 2.3: “and 2 alfalfa plant samples (each
sample being a pool of leaves from 10 individuals)” I think these two
points are not coherent.

=> The sentence on 2.1 was corrected as follow: "Alfalfa leaves were randomly collected from plants located in the crop field and in the neighboring meadow."